# Monitoring Fish Freshness in Real Time under Realistic Conditions through a Single Metal Oxide Gas Sensor

**DOI:** 10.3390/s22155888

**Published:** 2022-08-06

**Authors:** Giulia Zambotti, Rosamaria Capuano, Valentina Pasqualetti, Matteo Soprani, Emanuela Gobbi, Corrado Di Natale, Andrea Ponzoni

**Affiliations:** 1Unit of Brescia, National Institute of Optics (CNR-INO), National Research Council, Via Branze 45, 25123 Brescia, Italy; 2Unit of Lecco, National Institute of Optics (CNR-INO), National Research Council, Via Previati 1/E, 23900 Lecco, Italy; 3Department of Information Engineering, University of Brescia, Via Branze 38, 25123 Brescia, Italy; 4Department of Electronic Engineering, University of Rome Tor Vergata, Via del Politecnico 1, 00133 Rome, Italy; 5Agri-Food and Environmental Microbiology Platform (PiMiAA), Department of Molecular and Translational Medicine, University of Brescia, Viale Europa 11, 25123 Brescia, Italy

**Keywords:** gas sensor, fish, gas chromatography–mass spectrometry, microbiological count, metal oxide, temperature modulation

## Abstract

The realization of an unobtrusive and effective technology able to track fish freshness in real time and inform on its edibility is highly demanded, but still unachieved. In the present paper, we address this issue through a single metal oxide gas sensor working in temperature modulation mode. The system can work without an external reference air source, which is an appealing feature for its possible integration in domestic refrigerators. Tests were carried out using fresh sea bream fillets as case study and working both inside the refrigerator and at room temperature. Parallel gas chromatography–mass spectrometry and microbiological characterization indicated the marked dependence of both the microbiological condition and the gas-phase composition from the individual sample and from the storage temperature. Despite such a large variability, which may be expected in real applications, the proposed system provided similar responses whenever the total bacterial population approached and exceeded the edibility threshold of 10^7^ CFU/g.

## 1. Introduction

Foodborne diseases of microbiological origin can be caused by a variety of agents that gain entry via the gastrointestinal tract. Symptoms of foodborne disease, which are not necessarily confined to diarrhea and vomiting, are caused by viable organisms and/or by the toxins that they produce. The presence of foodborne agents that may cause illness poses a significant risk to consumer health. They may originate from poor practices, such as poor quality of raw materials or food components, undercooking, cross-contamination, poor cleaning, poor temperature, and poor time control [1]. A survey conducted in 2017 in 27 EU Member States (MS) reported 5079 food- and waterborne outbreaks, corresponding to an average number of 97.7 outbreaks per week within the EU. The number of cases due to fish and fishery products was higher than in 2016 (12,215 in 2017, 11,191 in 2016) and most of them, 42.5%, were hospitalized [2]. Traditional control tests used before product commercialization include visible examination (color, texture, odor, etc.), count of microorganisms with classical techniques (counts in plates, chemical methods, optical spectroscopy, enzymatic methods, etc.), and identification of microorganisms by metabolites (gas/liquid chromatography, etc.). All these screening tests are very efficient, but they carry high usage costs, a long time for analysis and processing the results, and well-trained staff. An efficient and portable technology, being at the same time unobtrusive and easy to implement in practical situations, is highly desirable, but still a challenge. The need for such a technology has been further emphasized by the progress of Internet of Things (IoT) applications [3]. Small sensing systems, able to continuously monitor the quality status of foods, providing a quick alert in cases of unhealthy conditions occurs are highly demanded for integration, wherever fresh foods are stored, processed, or transported, as in smart home refrigerators or in the industrial food chain [4].

Gas sensors represent an appealing solution, owing to their intrinsic low cost, reduced size, and power consumption characteristics merged with no need for sample preparation, which allows for easy and fast spoilage detection. Focusing in particular on fish, several papers have reported a correlation between spoilage and release of volatile basic nitrogen compounds (TVB-N) in the gas phase as a consequence of bacterial metabolism [5]. Based on these findings, different individual sensors featuring marked selectivity for TVB-N have been proposed, exploiting polymers [6,7], metal oxides (MOX) [8,9], hydrogels [10], dyes [11], and cellulose [12] as the sensitive material. Still, besides the promising performances reported for these systems, some issues remain. For example, dyes and hydrogels feature reversibility issues that hinder their continuous use [10,11], cellulose is typically exploited as an extremely cheap and disposable sensor and is more suitable for the development of smart packaging, rather than smart refrigerators, where the need for continuous use over long periods may represent a challenge for this natural material. Polymers and metal oxides are more promising candidates for this latter application, owing to their suitability for long-term operation and their reversible reactions with gases [6,7,13,14]. In the last few years, several research groups have worked on these materials to improve their capacity to detect such chemicals as triethylamine (TEA) or dimethylamine (DMA), considered markers of the TVB-N family. To mention a few examples, remarkable responses to a few ppm of TEA have been reported in the literature, together with promising partial selectivity against alcohols and ketones, with the response to TEA being about three to six times larger than the response to these compounds [15,16]. Similar results have also been reported for DMA detection [9]. Very recently, for both TEA and DMA, partial selectivity with responses about 10–20 times larger than the responses to alcohols and ketones has been reported, together with limits of detection (LOD) at sub-ppb and sub-ppm level, respectively, for TEA and DMA [13,17].

These results are quite promising, but have been obtained exposing sensors to individual compounds, not to gas mixtures. Confirming such potential with real samples in real conditions and proving the correlation of the sensor signal with gold-standard characterization are not trivial tasks. Difficulties arise from the complex composition of spoilage off-odors, which comprise a broad range of compounds, including, for example, alcohols, esters, ketones, and sulfides, in addition to TVB-N [18]. The simultaneous exposure to complex gas blends may strongly alter the sensor response with respect to the case of exposure to individual compounds [19,20]. Moreover, additional complications may arise from variations in environmental parameters that may occur in real scenarios and affect the volatility of compounds and the sensor performance [3]. To address selectivity within these complex situations, sensor arrays instead of individual sensors are often considered, though this means increasing the size and the cost of the sensing systems [21,22,23].

The capability of these devices, often called “electronic noses” (ENs), in the detection of microorganisms is mainly related to variations in the volatile compounds, the so-called olfactory fingerprint, of food products caused by microbial metabolism. ENs have been successfully applied in process monitoring [24,25], freshness evaluation [26,27,28,29], shelf-life examination [29,30,31], and authenticity and traceability determination [25]. Routine control of food contamination by ENs is one of its most promising applications. Despite its potential, two issues should be solved to further develop this technology at a level compatible with its integration in our daily life.

First, though ENs are portable, their size is rarely compatible with integration in domestic appliances, such as refrigerators. Second, the typical EN measurement is a comparative measurement using a stable reference air, which is generated from certified air bottles [30,31] or using ambient air filtered through a humidity-compensation system [32].

A solution with the potential to overcome these limitations is the use of MOX gas sensors working according to temperature modulation, instead of at constant temperature (as in sensor arrays). In particular, temperature modulation results in a periodic modulation of sensor electrical resistance with time, whose gas-dependent shape can be used as input for pattern-recognition software [33,34]. In principle, temperature profiles allow for adopting the same approach of the traditional EN, but using a single sensor instead of an array [33,34], hence reducing size and power consumption. Moreover, the sensing system can work without any reference air, since every atmosphere has its own curved fingerprint.

Within this framework, an interesting work was carried out by S. Marco and coworkers, who developed a system based on a single metal oxide gas sensor working with temperature modulation and tested it with fish samples inside a refrigerator [35]. This approach is extremely appealing and the paper was inspiring, however, some important aspects should be further investigated to properly develop and demonstrate the effectiveness of the proposed approach in real conditions. These include, for example, the investigation of a wider range of practical conditions that can be encountered in real scenarios and comparison with certified techniques to verify the effective correlation between the sensor signal and the food properties.

To progress in the development of this type of technology, in this article we propose an EN-like system based on a commercial micromachined MOX sensor exploiting temperature modulation to track the degradation of fish in different environmental conditions, using sea bream as a case study. This sensing system is extremely small and can be easily implemented within such appliances as domestic refrigerators. The suitability of the proposed system is validated by parallel microbiological characterization of sea bream fish samples and the chemical analysis of the developed headspace. Given the strong temperature dependence of the bacterial metabolism and related off-odors, different environmental temperatures are used to simulate conditions that may be encountered in real situations and that may challenge the selectivity of the device and, in turn, its capability to properly identify inedibility.

## 2. Materials and Methods

### 2.1. Tested Samples

Fresh sea bream filets packed in trays were bought at a local supermarket on different days. For each sample, we took care to start the related experiments the day after of the packaging day indicated on the commercial label. Four fish packages were tested, 2 for each analyzed temperature. For each tray, the content was divided into 10 g portions and all portions were individually stored at the conditions of the planned experiment (Figure 1), namely, inside the refrigerator at 4 °C or at room temperature (25 °C).

For gas sensing tests, the 10 g portion was put into a beaker (100 mL volume) and sealed with parafilm, leaving a small hole for insertion of the sensing system. This was kept close to the fish sample for the whole duration of the experiment to track continuously the time evolution of the developed headspace. An example of the measurement setup is provided in Figure 1a.

Timing was scheduled as follows: each sample was put inside the beaker (at time = 0 h, hereafter T0) and incubated for 24 h (hereafter T24) for fish kept at 25 °C or for 32 h (T32) for fish at 4 °C. The incubation time was chosen as a reasonable time to allow bacteria population growth until the edibility threshold had been surpassed. Since the bacterial growth rate decreases with decreasing incubation temperature, we kept samples monitored for a longer time in experiments carried out at refrigerator temperature.

Reference samples, namely, air and sterilized water, were tested with the same experimental setup, i.e., using an empty beaker filled with ambient air and a beaker containing 10 mL of distilled water, respectively.

The tested samples are summarized in Table 1, which describes storage and testing conditions and the initial and final total viable counts (TVC). Please refer to Section 2.3 for details about TVC.

### 2.2. Sensing System

A miniaturized commercial platform (JLM Minimox from JLM innovation Gmbh, Tübingen, Germany) hosting a micromachined MOX sensor (AMS CCS801 from AMS-OSRAM AG Premstaetten, Austria) was used in this study. A picture of the device is shown in Figure 1b. The JLM Minimox is a plug-and-play system with hardware and software to control the signal of the hosted sensor and its temperature through a voltage heater. The AMS CCS801 sensor is an aspecific gas sensor nominally sold for the general purpose of air-quality monitoring [36,37]. The custom modulation of temperature was applied to make it more selective and suitable for the goal.

The temperature profile was controlled by applying to the heater a squared voltage wave for a period of 20 s, consisting of a constant voltage V_HEATER_ = 1.65 V applied for 10 s and then a constant voltage of V_HEATER_ = 2.31 V for another 10 s. The sampling time for the sensor resistance was set to 0.2 s in order to properly track the resistance vs. time curve.

Figure 2 shows an example of the sensor resistance variation with time acquired during the experiments carried out at 4 °C, with the sensor exposed to the environmental air, Figure 2a, or to the headspace generated by a fish sample after 14 h, Figure 2b. The voltage bias applied to the heater is also shown (light-blue line in the graph). The sensor resistance changed with time due to the modulated heater stimulus but also depending on the surrounding gas phase. Every thermal period, the values spanned by R_SENSOR_ and the shape of its time dependence were resumed through a set of parameters.

Differently from common ENs exploiting sensor arrays working at constant temperature, this type of measurement does not necessarily require reference air to calculate a differential response and allows almost continuous monitoring of the signal. The duration of the thermal period represents the characteristic time scale of this sensing system. For those applications in which the gas phase changes slowly with respect to the thermal period (20 s in the present work), as is the case in food degradation, a set of parameters can be extrapolated every period to describe the time dependence of the headspace smelled. These parameters, hereafter called features, were calculated according to the following procedure.

With reference to Figure 2, for each thermal period, four resistance values were extrapolated:R_1_: the sensor resistance measured during the second sampling time in which V_HEATER_ is in the low state. Considering that the thermal time constants of microhotplate-based sensors are of the order of a few milliseconds [38], a single sampling time (0.2 s) allowed for reaching a steady-state temperature.R_2_: the sensor resistance measured 2 s after the V_HEATER_ was switched from the high to the low state.R_3_: the sensor resistance measured during the last sampling time in which V_HEATER_ was kept in the low state.R_4_: the sensor resistance during the second sampling time in which the V_HEATER_ was in the high state.

Based on these R_SENSOR_ values, the following features were calculated:
Delta-C = variation in R_SENSOR_ measurements during the cold half-period, i.e., Delta-C = R_1_ − R_3_;Delta-CH = variation in R_SENSOR_ measurements between the end of the cold half-period and the beginning of the next hot half-period, i.e., Delta-CH = R_3_ − R_4_;Ratio-CH = ratio between the R_SENSOR_ measurements at the end of the cold half-period and at the beginning of the next hot half-period, i.e., Delta-CH = R_3_/R_4_;Slope-C = slope of the R_SENSOR_ vs. time curve extrapolated during the initial stage of the cold half-period, i.e., Slope-C = (R_1_ − R_2_)/(t_2_ − t_1_), where the time interval is t_2_ − t1 = 2 s.

These features were extrapolated for every thermal period, i.e., every 20 s, for the whole duration of the experiment and further processed calculating the average values over six measurements, corresponding to an average feature-value extrapolation every 2 min.

We used principal component analysis (PCA) to visualize the response of the sensing device and explored its correlation with the status of the samples.

The set of the adopted set of features was heterogeneous: it included resistance ratios, resistance differences, and slopes of resistance vs. time. This is useful to catch different characteristics of the sensor signal and their correlation with the samples’ properties. Considering the different scales of these features, we applied an autoscaling procedure before further processing [39]. Specifically, averaged features were individually autoscaled by subtracting their mean value and dividing by their standard deviation. The processed features were then used as inputs for the PCA function of MATLAB. In the score plots, we used a label matrix to characterize the status of each measurement/group of measurement. The labels used were:Sample, as described in Table 1.Temperature kept during sensing test, as listed in Table 1.Time, which refers to the testing time, starting from T0 as described in Section 2.1.Total viable count (TVC), which was carried out at discrete times according to the procedure described in Section 2.3. As better detailed in Section 2.3 and further discussed in Section 3, it is not possible to have a continuous TVC. This characterization should be done at discrete times, and so far only a few measurements have the corresponding TVC label.

Plotting of PCA scores was carried out with homemade algorithms inspired from the literature [20], which allowed for double labeling each datum to better visualize the relationship with labels describing the sample status.

### 2.3. Microbiological Characterization

In parallel to the EN tests, total viable counts were carried out in order to quantify the bacteria present. Counts was done using two portions of fillet (10 g), maintained in the same conditions as the samples used for EN tests of the fish samples. Fish subsamples were taken at different times during the experiment. Before performing the microbiological count, the subsamples were homogenized in a stomacher with physiological solution (90 mL). Subsequently, one aliquot (100 µL) from each sample was serially diluted and each dilution surface plated in duplicate over plate count agar, a nonselective medium for the growth of microorganisms. Plates were incubated at 30 °C for 48 h, then the colonies grown were counted to provide the real concentration of bacteria in the samples (mesophilic counts).

### 2.4. Chemical Characterization

GC/MS was used in order to evaluate the volatile compounds’ pattern evolution during fish degradation. Volatile organic compounds (VOCs) released by fish samples stored at 25 °C were collected at 0, 4, 8, 21, 23, and 25 h of incubation. Regarding fish stored at 4 °C, six separate samples were maintained at 4 °C and each analyzed only once: at 0, 4, 8, 21, 25, and 30 h of storage. VOC analysis were carried out at room temperature, i.e., at 25 °C. Volatile organic compounds composing fish headspace were preconcentrated using solid phase microextraction (SPME). The analysis was performed immediately after VOC preconcentration, using a GCMS-QP 2010 series (Shimadzu, Kyoto, Japan) gas chromatograph mass spectrometer.

A 50/30 μm divinylbenzene/carboxen/polydimethylsiloxne (DVB/CAR/PDMS-Supelco, Bellefonte, PA, USA) fiber was exposed in the fish headspace for 1 h and 30 min at room temperature (T = 25 ± 2 °C). VOCs were subsequently desorbed in the GC injector at 250 °C for 5 min in splitless mode. Analysis was performed immediately after VOC preconcentration, using the gas chromatograph mass spectrometer. Chromatographic separation was performed using an EQUITY-5 capillary column (poly 5% diphenyl/95% dimethyl siloxane) phase (Supelco, Bellefonte, PA, USA), with the following sizes: 30 m length × 0.25 mm internal diameter × 0.25 μm film thickness. Starting from 40 °C (held for 7 min), the oven temperature was raised to 230 °C (held for 10 min) at an increasing rate of 15 °C/min, then, using a ramp increase rate of 20 °C/min, it reached 300 °C, and this temperature has been held for 1 min. Ultrahigh-purity helium was used as carrier gas, maintaining linear velocity constant mode at 27.7 cm/s.

The carrier pressure was 14.5 kPa, working with a total flow of 5.4 mL/min and a column flow of 0.59 mL/min. The mass spectrometer was a single quadrupole analyzer operating in electron ionization mode (EI) at an energy of 70 eV. Transfer line and ion source temperature was maintained constant at 250 °C. Data were acquired in the full-scan mode, scanning over an *m/z* mass range between 20 and 450 amu (atomic mass units). Putative compound identification was performed using both NIST 127 and NIST 147 libraries.

## 3. Results and Discussion

### 3.1. Headspace Analysis

Fish contain small amounts of carbohydrates and typically have a high content of free amino acids and proteins. According to the literature, the major cause of fish spoilage is microbial growth and associated metabolism, which cause the formation of several compounds, including amines, sulfides, alcohols, aldehydes, ketones, and organic acids with unpleasant off-flavors [5]. Considering the strong temperature dependence of both the bacterial metabolism and the volatility of molecules, the composition of the volatilome was analyzed at the two target temperatures of 4 °C and 25 °C. SPME coupled with GC/MS analysis allowed detection of a total of 86 compounds. Most of them presented fluctuations during the storage time or appeared sporadically in some samples. However, 13 VOCs were correlated with the fish shelf-life (see Table 2). For the analysis, only compounds with similarity scores greater than 80% with the reference mass spectrum were considered.

Seven VOCs were present in the headspace of samples stored in both conditions. Figure 3 shows the concentration trends of these VOCs.

Differently, six VOCs were detected only in the headspace of fish samples stored at room temperature, specifically, dimethyl disulfide, 2,3-heptanedione ethyl crotonate, ethyl tiglate, dimethyl trisulfide, and indole (Figure 4).

Ethyl tiglate was detected after 8 h of fish storage, and 2,3-heptanedione, ethyl crotonate, and dimethyl trisulfide were first identified in sample headspace after 21 h of storage at room temperature. For ethyl crotonate and dimethyl trisulfide, the concentration registered a rapid increase in the next 2 h and then remained stable until 25 h, while for the 2,3-heptanedione, the maximum concentration was reached at 21 h and a slight decrease recorded in the following hours. Dimethyl disulfide and indole were first detected only after 23 h in the sample maintained at room temperature. Dimethyl disulfide concentration remained quite stable in the 2 consecutive hours, while indole increased. The detected compounds were in fine agreement with literature reporting headspace analysis of fish and seafood products (e.g., sea bass, sea bream, mackerel, cod, and whiting shellfish) stored in aerobic conditions. They are usually associated with microbiological spoilage, chemical oxidation of fatty acids, or enzymatic activities [40,41,42,43,44,45,46]. Almost all those compounds show an increased amount during storage, both for fish stored in the refrigerator and at room temperature, according to the literature. Detection of alcohols and ketones has been reported to be associated with metabolic activity of some microorganisms responsible for food spoilage, such as *Pseudomonas* spp., *Shewanella* spp., *Enterobacteriaceae*, *Brochothrix thermosphacta*, lactic acid bacteria, and *Photobacterium phosphoreum* [46,47,48]. Ethyl esters (e.g., ethyl crotonate) and sulfur-containing compounds are usually released as product of *Pseudomonas* spp. activity and are usually associated with seafood deterioration [49,50,51]. Some papers [42,48] have reported an increase in indole during storage. This compound is usually released in the presence of *Escherichia coli* and *Pseudomonas* spp. food spoilage. In particular, at high temperatures, significant production of indole is strictly correlated with *E. coli* levels [52]. These results highlight the complexity of the volatilome developed during the spoilage process and its marked dependence on storage time and temperature.

### 3.2. Microbiological Analysis

The complete degradation of fish was reached at different incubation times depending on the temperature, as evidenced by the microbiological characterization. To correlate the sensor responses with the microbiological status of the fish samples, microbiological analysis of the samples was carried out in parallel with the sensor measurements. In particular, the TVC was chosen, owing to its strict relationship with the threshold limits of food edibility [1].

TVC results are shown in Figure 5. Notably, the starting conditions (TVC value at time t = 0 h) for each sample are compatible with the edibility threshold, typically reported as TVC < 10^7^ CFU/g, [1]. Nonetheless, the specific TVC values strongly varied from sample to sample. TVC differences may be expected in commercial samples bought on different days at a store: each portion may come from a different fish with respect to other portions; moreover, variations may occur in the storage and cutting processes, as well as in the preparation conditions. Most samples exhibited an exponential growth in TVC, with higher temperatures inducing a faster growth rate, in agreement with typical microbial behavior. An exception is the Fish1 sample, which showed a surprisingly low initial microbiological population (TVC < 10^5^ CFU/g) that remained at nearly the same level for a period of 24 h, even if kept at 25 °C. Coherently, an inspection by the human nose did not reveal any unpleasant odor, which instead characterized the other fish samples after a period of 24 h. We may reasonably speculate that this sample, during its food-chain preparation, got exposed to some antimicrobial action/agent.

### 3.3. Sensing System Characterization

The responses of the sensing system are reported as PCA score plots in Figure 6 and Figure 7. The PCA shows that the 1st and the 2nd principal components represent about 50% and 47% of the total variance, respectively. Their sum (≅97%) indicates that the first two principal components account for most of the information detected by the sensing system, while the almost equal distribution between the two components can be reasonably ascribed to the complexity of the volatile information, which cannot be resumed through a single dominant component. This is in line with results discussed in Section 3.1, which revealed a heterogeneous headspace featuring a marked temperature and time dependence, and in Section 3.2, which showed the widespread microbiological status of individual samples. These arguments will be discussed in the following on the basis of the sensor signal.

As a preliminary consideration, the proposed system reveals good capacity to track the time evolution of the headspace generated by each given sample, owing to the intrinsic effectiveness of the system to perform measurements very rapidly, allowing extrapolation of responses every 2 min (see the experimental section for details). This is a clear advantage with respect to traditional ENs; indeed, the speed of these latter systems is limited by the response time of metal oxide gas sensors and by their recovery time, i.e., the time required by the sensor resistance to restore its baseline value once the reference air is restored in the sensor chamber. These times typically add up to the order of several minutes or even tens of minutes [29,53]. Figure 6 shows the comparison between data acquired with one representative sample for each category (air, water, and fish). (a) and (b) subplots show different information for the same samples/data. More specifically, in Figure 6a the sensor responses to the different samples are reported as a function of the storage temperature kept during the experiment; in Figure 6b, the sensor responses to the same samples are presented with reference to the time of their analysis.

As shown in Figure 6a, data related to air and water differ owing to the different absolute humidity (AH) concentration, but they remain in a restricted region for the whole observation time. On the contrary, data recorded with fish samples evolve with time as a consequence of the time dependence of their volatilome and bacterial population, reflecting the results observed through the GC/MS and microbiological analysis. The marked temperature dependence of the gas-phase content is also evident from the sensor signal in the PCA score plots. Indeed, as better shown in Figure 6b, the odor released by the sample stored at 25 °C is tracked in the PCA plot by data points moving vertically with time along the 2nd component from the center-right part of the plot to the bottom-right corner. The signal recorded with the sample stored in the refrigerator moves in a diagonal direction from the center-left portion of the plot to the bottom-right corner, where it overlaps with the other fish sample.

The relationship with the microbiological count is reported in Figure 7, showing all the fish samples analyzed in this work (see Figure 1 and Figure 5).

To correlate the sensor signal and the TVC results, it is worth stressing that while the sensor is able to provide an almost continuous response with time, the same is not true for the microbiological characterization, which is carried out at discrete times. To match the two datasets, the sensor responses shown in Figure 7b were subsampled, reporting four averaged measurements (corresponding to a period of 8 min) for each time corresponding to the TVC analysis. Eight minutes is a short time compared to the time growth of microbiota population. During this time, we can reasonably consider the bacterial metabolism and the measured TVC value as “static,” providing a reliable comparison between the two techniques.

As already introduced in the discussion of Figure 7, fish samples were strongly distinguished by the storage temperature, especially at the early stages of experiments. Their data points in the PCA plot follow a vertical path remaining on the right side of the plot for experiments carried out at 25 °C and a diagonal path from center-left to bottom-right for experiments carried out in the refrigerator. This is not surprising considering the strong temperature effects also observed with the GC-MS and microbiological characterization. Indeed, in the gas-phase analysis, differences emerge not only concerning the relative abundance of the same compounds, but some compounds were found only in the 25 °C samples and were absent in the refrigerated condition. For example, this is the case of sulfide compounds, such as dimethyl disulfide and dimethyl trisulfide reported in Figure 4, which are often reported to induce strong responses in metal oxides [54,55]. Moreover, according to the TVC parameter (Figure 5), samples differ from one another, and this may support the separation in the PCA plot observed also between samples stored in the same conditions, especially during the first hours of the experiment.

Despite these differences, Fish 2, 3, and 4 are all characterized by the typical exponential growth of the TVC (Figure 5) and their status changes with time from edible (TVC < 10^7^ CFU/g) to not (TVC ≥ 10^7^ CFU/g). It is interesting to observe that, despite the signals of the fish samples held at 25 °C and 4 °C being quite different at the beginning, in the PCA score plot the signals of these samples converge toward the same area as time and TVC increase. For practical applications, this is an interesting result.

Storage temperature may vary from refrigerator to refrigerator, also depending on customer settings and preferences, and drifts in the internal system dedicated to temperature control may also occur, owing to seasonal variations and/or long-term operation. Considering these possible situations, it would be desirable to have a sensing system able to accomplish its task independently of the storage temperature. Moreover, it is worth noting that concerning Fish1, which is characterized by TVC remaining very low for the whole observation time, the sensor response never approached the aforementioned corner. Though the uncertainty related to the history of this sample does allow us to clearly explain why its TVC remained low over time (see Figure 5 and related discussion), the coherence between the microbiological and the sensing characterizations represents a further confirmation of the capability of the proposed sensing system to identify the edibility status of fish.

## 4. Conclusions

In this paper, we have presented a sensing system suitable to determine fish edibility through the monitoring of the developed off-odors. The system is based on a single metal oxide gas sensor working with temperature modulation, which allowed us to achieve the required selectivity by means of an unobtrusive, USB-key-sized device, with no need for any air or gas reference source. Choosing fresh sea bream as the target sample and working at two temperatures, namely, 4 °C and 25 °C, which may be commonly encountered in real scenarios, we demonstrated the effectiveness of the proposed system to detect inedibility.

Indeed, the parallel microbiological and GC/MS analyses revealed large variability for the bacterial population of different samples and marked temperature effects on the released volatilome. Nonetheless, as the microbial population exceeded the edibility threshold of 10^7^ CFU/g, the response of the sensing system systematically converged to the same region in the PCA score plot. These are promising results for the development of sensing systems suitable for integration into smart domestic refrigerators and/or in industrial plants. In future work, we plan to progress in the development of this technology by investigating its performance in more complex scenarios, including the addition of foods other than fish that may be present inside the refrigerator and may release interfering odors. The goal is to approach step by step the complexity of situations that may be encountered in real appliances.

## Figures and Tables

**Figure 1 sensors-22-05888-f001:**
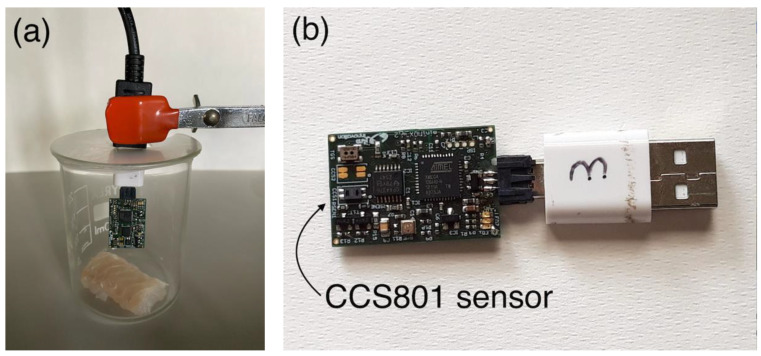
Measurement setup (**a**). The sensing system is lodged inside a beaker containing the target sample. Picture of the sensing system device (**b**).

**Figure 2 sensors-22-05888-f002:**
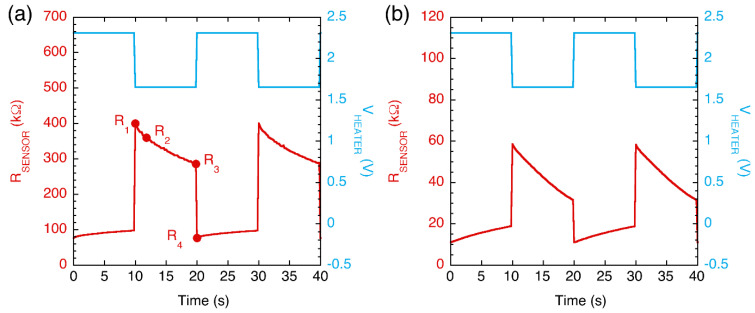
Sensor resistance vs. time curves acquired during sensor exposure to air (**a**) and to the headspace of a fish sample at the storage time of 14 h (**b**). In both cases, experiments were carried out inside the refrigerator (4 °C). The modulation of the heater voltage is also shown. Full circles in (**a**) indicate the resistance values used to extrapolate parameters (features) that resume the shape of the curve (see text for details).

**Figure 3 sensors-22-05888-f003:**
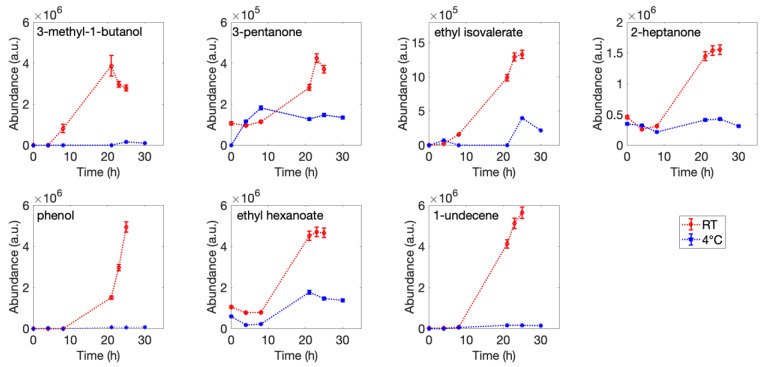
SPME-GC/MS analysis of fish samples stored at 4 °C and 25 °C. Time dependence of the abundance of the most meaningful compounds (see text for details) present in the headspace for fish stored at room temperature (25 °C) and inside the fridge (4 °C).

**Figure 4 sensors-22-05888-f004:**
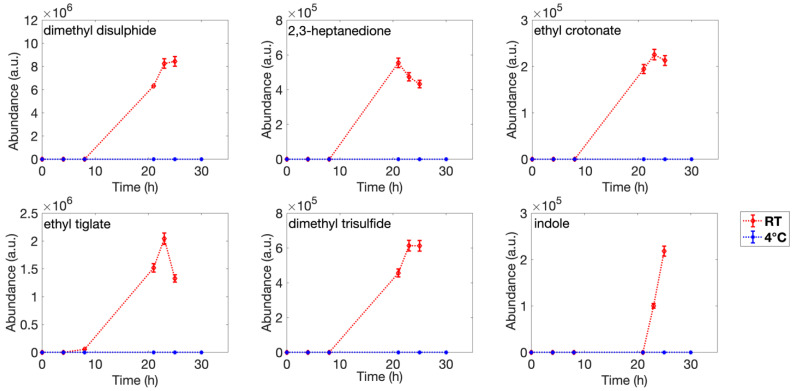
SPME-GC/MS analysis of meaningful compounds found only in the headspace of fish stored at room temperature (25 °C), related to storage time.

**Figure 5 sensors-22-05888-f005:**
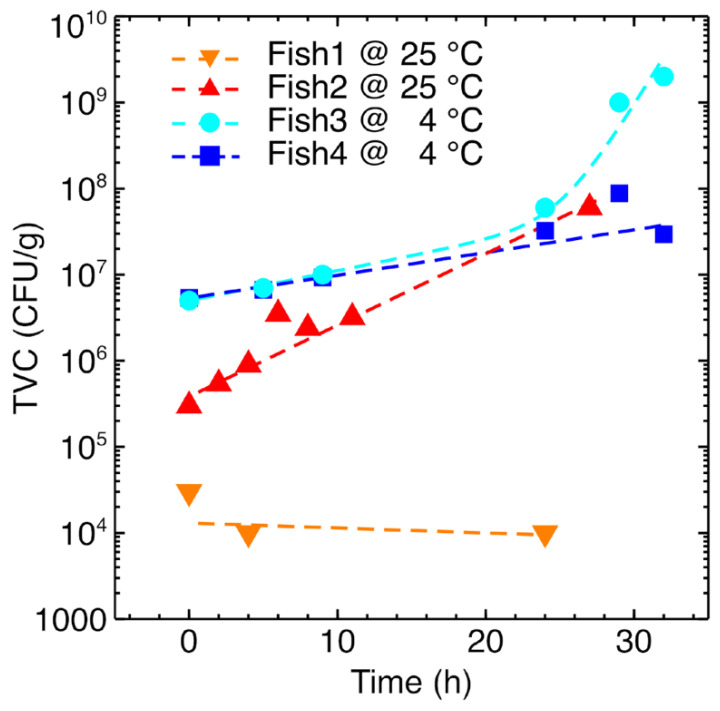
Microbiological characterization of fish samples: time dependence of the total viable count (TVC). Dashed lines are guides for the eye.

**Figure 6 sensors-22-05888-f006:**
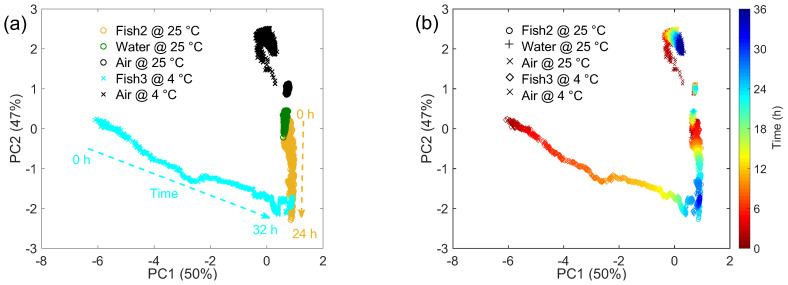
Principal component analysis (PCA) plot showing the time dependence of the device response and its correlation with selected air, water, and fish samples measured at different temperatures. For the sake of clarity, only five samples are shown; each sample is measured for a period lasting between 24 and 36 h. (**a**) Colors and symbols are used to distinguish different samples and temperatures, respectively. Dashed arrows indicate the time direction for fish samples. (**b**) The color-bar identifies the measurement time for all samples, symbols identify different samples.

**Figure 7 sensors-22-05888-f007:**
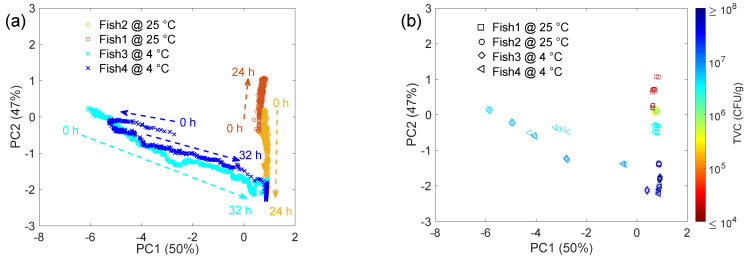
Principal component analysis (PCA) plot showing the correlation between the sensor signal and the microbiological characterization (TVC). (**a**) Full dataset for fish samples, dashed arrows indicate the time direction for each sample. (**b**) Subsample data (see text for details) showing the correlation between the score plot and the respective total viable count (TVC).

**Table 1 sensors-22-05888-t001:** Tested samples and main parameters of status, namely, the storage and testing conditions and the initial and final total viable counts (TVC). The TVC values were determined following the procedure described in Section 2.3.

Sample	Initial TVC (CFU/g)	Final TVC (CFU/g)	Storage and Testing Condition
Fish1	1 × 10^4^	3 × 10^4^ @ 24 h	Thermostatic chamber @ 25 °C
Fish2	3 × 10^5^	6 × 10^7^ @ 24 h	Thermostatic chamber @ 25 °C
Fish3	5 × 10^6^	2 × 10^9^ @ 32 h	Refrigerator @ 4 °C
Fish4	5 × 10^6^	3 × 10^7^ @ 32 h	Refrigerator @ 4 °C
Sterilized Water	<1	<1 @ 32 h	Thermostatic chamber @ 25 °C
Air	Not performed	Not performed	Thermostatic chamber @ 25 °C
Air	Not performed	Not performed	Refrigerator @ 4 °C

**Table 2 sensors-22-05888-t002:** SPME-GC/MS analysis of fish samples’ headspace. List of volatile organic compounds (VOCs) most correlated with sample degradation. For each VOC, the retention time, the CAS number (°# CAS) and the reference mass spectra similarity are specified. In the Samples column, R indicates samples stored and analyzed at 25 °C, while F indicates samples stored at 4 °C and analyzed at room temperature (25 °C).

Number	Retention Time (min)	Putative Identification	°# CAS	Reference Mass Spectra Similarity (%)	Samples
1	5.160	3-methyl-1-butanol	123-51-3	92	R; F
2	5.306	dimethyl disulfide	624-92-0	94	R
3	7.293	3-pentanone	96-22-0	80	R; F
4	8.927	2,3-heptanedione	96-04-8	91	R
5	9.259	ethyl crotonate	10544-63-5	95	R
6	9.626	ethyl isovalerate	108-64-5	92	R; F
7	10.627	2-heptanone	110-43-0	88	R; F
8	11.987	ethyl tiglate	5837-78-5	93	R
9	12.555	dimethyl trisulfide	3658-80-8	91	R
10	13.024	phenol	108-95-2	96	R; F
11	13.340	ethyl hexanoate	123-66-0	97	R; F
12	15.153	1-undecene	821-95-4	91	R; F
13	18.611	indole	120-72-9	87	R

## Data Availability

The data presented in this study are available upon reasonable request from the authors.

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
