# Peer review of "Monitoring Fish Freshness in Real Time under Realistic Conditions through a Single Metal Oxide Gas Sensor"

_sensors, 2022, doi:10.3390/s22155888_

Round 1

Reviewer 1 Report

Zambotti et al. report a single metal oxide gas sensor for the monitoring of freshness in real time under realistic condition. The authors fabricated a single metal oxide gas sensor working in temperature modulation mode.

Although the authors’ study is a necessary approach in food industry especially for the monitoring of freshness and their system possesses potential advantages, in the aspect of the biosensing application, but there are fundamentally unclear on the novelty and scientific advancement of this method.

We could recommend revision for the following reasons: there are fundamentally unclear on the novelty and scientific advancement of this method. The authors designed and fabricated a single metal oxide gas sensor that works in realistic condition without an external air source. Real time test is performed using fresh seabream fillets as a sample in different temperature conditions.  

However, this approach is very similar to the previous study (Perera, A., Pardo, A., Barrettino, D., Hierlermann, A., & Marco, S. 2010, 146(2), 477-482). Though the real time conditions for temperature modulation is differed, but it is also difficult to being accepted as an advancement. Thus, the novelty and scientific advancement of this method is unclear. The authors need to add the novelty and scientific advancement in manuscripts.

In addition, the manuscript requires additional editing for the figures. In some pictures, including picture 3 and 4, is not clearly seen and some of the focus is not clearly expressed.

Reviewer 2 Report

In this manuscript, the authors Zambotti et. al developed a single metal oxide gas sensor to track the fish freshness in real time. They used fresh seabream fillets as case study and working both inside the refrigerator and at room temperature. The results of parallel gas chromatography - mass spectrometry and microbiological characterization indicated the marked dependence of both the microbiological condition and the gas-phase composition from the individual sample and from the storage temperature. This work could be accepted after minor revisions. Other questions were shown below:

1. Metal oxide gas sensors that can detect volatile basic nitrogen compounds have been very numerous in recent years and show excellent performance. But in the introduction of this manuscript, only the poor selectivity of metal oxide gas sensors is highlighted, and I think that some examples can be given to illustrate the advantages and research progress of metal oxide gas sensors. And by contrast, it is very important to highlight the innovation point of this manuscript.

2. The meaning of R1-R4 in Figure 2 should be clearly explained.

3. The case of words in the table should be unified, such as the “2-heptanone” in Table 2. At the same time, the font of text between different tables should be kept unified.

4. Please add a physical photograph of the sensor and indicate the principle of their testing with a circuit diagram.

5. Which metal oxide is selected? Please explain the reasons for your choice.

6. Can other interfering gases in the refrigerator have an effect on the sensor? For example, meat spoilage also produces H2S, please specify experimentally.

Reviewer 3 Report

The work is devoted to the development of an accessible and compact method for determining the freshness of fish, which is undoubtedly an important and urgent task. The solution proposed by the authors in the future can be integrated into storage systems and significantly improve food security. The results of the work are interesting and are confirmed by a set of methods. The work can be recommended for publication, but after fixing of some issues. Comments are below:

1.     In section 2.2, the authors write that 4 parameters were calculated from the experimental data: Delta-C, Delta-CH, Ratio-CH, Slope-C. The further transition to the principal component analysis (PCA) is described very briefly, and most likely will not be understood by a wide audience of readers. Since reproducibility is one of the key criteria for scientific publication, the reviewer believes that the authors should describe the analysis in more detail, especially since they performed it using "home-made algorithms" (line 179).

2.     Question related to previous comment. In line 178 authors say: “Averaged features were then individually auto scaled (subtracting their mean value and dividing by their standard deviation).” Does that mean that data processing can be done only when the experiment is completed? That is, in order to understand that the fish is edible at a given time, we need to wait until it deteriorates and the mean value with standard deviation will become known? What then is the point of such a system?

3.     What is the reason for the choice of incubation time, and why is it different for 4 and 25 C?

Minor issues

4.     Figure 1 and the table describing the experimental conditions should be separated. The table must be prepared using the text editor, not pasted as a bitmap.

5.     Table 1 should be arranged in accordance with the requirements of the journal (see template).

6.     Plots in Figures 3 and 4 have a very low resolution and are very hard to read. Replace with better images, as at least in the case of Figure 5 (although it could also be higher resolution).

7.     “Pseudomonas” (line 287) should be italic.

Round 2

Reviewer 1 Report

The authors revised well about the revision of the manuscript. The authors added information for the novelty and scientific advancement of manuscripts.  Also, method section have added the fresh sea bream filets as a sample in different temperature condition. In addition, figure 3 and 4 have revised clear image. However,  axis information of Figure 3 and 4  have not shown. Thus, authors need to add the axis information of image of Figure 3 and 4.  

Reviewer 3 Report

The authors fixed most of the issues, and the manuscript is perceived much better. However, apparently there was a technical error when inserting Figures 3 and 4. Now they do not have any captions, i.е. it's just a set of some graphs without marked axes and names of substances. Please correct.
